# Succinylation Modified Ovalbumin: Structural, Interfacial, and Functional Properties

**DOI:** 10.3390/foods11182724

**Published:** 2022-09-06

**Authors:** Gan Hu, Jiemin Zhang, Qi Wang, Meihu Ma, Lulu Ma, Shugang Li

**Affiliations:** 1National Research and Development Center for Egg Processing, College of Food Science and Technology, Huazhong Agricultural University, Wuhan 430070, China; 2School of Food and Biological Engineering, Hefei University of Technology, Hefei 230009, China

**Keywords:** ovalbumin, succinylation, physicochemical property, structure, functional property

## Abstract

In this study, ovalbumin (OVA) was succinylated with the addition of different levels of succinic anhydride, and the structural and functional properties of succinylated OVA (SOVA) were investigated. SDS−PAGE and FTIR spectrum confirmed the covalent attachment of the succinyl group to OVA. Thermal stability and the absolute value of zeta potential (pH 6.0) of SOVA were enhanced by 14.90% and 76.77% higher than that of the native OVA (NOVA), respectively. Circular dichroism (CD) spectra demonstrated that the succinylation decreased the α−helix and increased β−sheet content to 21.31% and 43.28%, respectively. The content of free sulfhydryl groups increased and intrinsic fluorescence spectra suggested the SOVA became more unfolded and flexible as the degree of succinylation enhanced. Furthermore, succinylation effectively enhanced the solubility and decreased the interface tension (oil−water and air−water interface) of OVA. Compared to NOVA, the emulsifying activity and stability of SOVA were increased by 1.6 times and 1.2 times, respectively, and foaming capacity and stability were enhanced by 2.7 times and 1.5 times, respectively.

## 1. Introduction

Egg whites are widely used in the food industry as a cost-effective and popular source of high–quality protein [1]. Ovalbumin (OVA), the most abundant protein in egg whites, accounting for more than 50% of the total protein in egg whites, is an important source of animal protein to provide a variety of essential amino acids. OVA is a typical globulin with a relative molecular mass of 45 kDa and an isoelectric point of 4.5. It consists of 385 amino acid residues, of which more than 50% are hydrophobic amino acids [2]. The functional properties of OVA, including emulsification properties, foaming, and gelation, have important applications in improving the taste and texture of food products [3]. However, the applications of OVA suffer from its dense globular structure, low molecular flexibility, and poor adsorption behavior at oil–water or air–water interfaces, which are related to the functional properties and can be improved by modifying the structure and aggregation state with appropriate methods.

Chemical modifications to the structure and function of proteins have shown advantages in high controllability, fast reaction, and mass production [4]. The current chemical modifications of proteins are mainly focused on glycosylation, phosphorylation, and acylation, by introducing groups to change the structure, the hydrophobic groups, and the electrostatic charge on the surface [5], where acylation is of great interest because of its high efficiency.

Succinylation is a type of typical acylation modification to increase the net surface charge and spatial resistance of protein by introducing a negatively charged succinamide group by covalently binding succinic anhydride to the nucleophilic groups. This modification technique is mild and effective that can significantly modulate the charge density on the surface of proteins [6]. In addition, succinic anhydride, commonly used as an acylating agent, is considered generally safe by FDA (Food and Drug Administration) [7]. Numerous studies have shown that succinylation can effectively improve the solubility, foaming, emulsification, and gelation of different proteins, such as oat protein isolate [8], mung bean protein isolate [9], and chicken liver proteins [10]. The effects of succinylation on the physicochemical properties and functional properties of OVA are studied in this project.

In this study, we modified OVA using different concentrations of succinic anhydride and investigated the effects of different degrees of modification on the functional properties (solubility, foaming, and emulsification) of OVA. By analyzing the structural and physicochemical changes of OVA before and after modification, we explored the mechanism of improving the functional properties of succinylated proteins. The results of this study provide new methods of succinylation and new techniques for the development of OVA products.

## 2. Materials and Methods

### 2.1. Materials

OVA (purity ≥ 98%) was purchased from Shanghai Yuanye Bio-Technology Co., Ltd. (Shanghai, China). Succinic anhydride, 2,4,6–trinitrophene sulphonic acid (TNBS), glycine, sodium dodecyl sulfates (SDS), Coomassie Brilliant Blue R–250, 5,5–dithiobis (2–nitrobenzoic acid) (DTNB), 8–anilino–1–naphthalenesulfonic acid ammonium salt (ANS), ethylenediaminetetraacetic acid (EDTA), and potassium bromide (KBr) were acquired from Sigma–Aldrich Co. LLC (Shanghai, China). Albumin bovine (BSA, purity ≥98%) was obtained from BioFroxx (Einhausen, Germany). All other reagents used in this study were of analytical grade.

### 2.2. Synthesis of Succinylated Ovalbumin (SOVA)

The SOVA was prepared following our previous procedure [11]. Briefly, protein dispersion (2%, *w*/*v*) was obtained by fully dissolving OVA in deionized water. Then, a quantity of succinic anhydride was added to the protein dispersion for the succinylation reaction. During the reaction, the pH of the solution was stabilized at ~8.0 using 2 M NaOH. When pH was kept at pH 8.0 without further change for 1h, the reaction was terminated by adjusting the pH of the solution to 7.0 using 1 M HCl. The resulting solution was dialyzed for 48 h, with water being changed every 8 h (the dialysis bag had a cut–off molecular weight of 14 kDa), and the dialyzed solution was further lyophilized at −40 °C for 48 h to obtain SOVA where 5%, 10%, and 20% (*w/w*, protein basis) of succinic anhydride was added into OVA and named SOVA–1, SOVA–2, and SOVA–3, respectively. The native OVA without modifications was set as the control group and named NOVA.

### 2.3. Measurement of the Degree of Succinylation

The level of modification was determined by measuring the free amino group using the TNBS method [12]. Briefly, the protein dispersions (1%, *w*/*v*) and the TNBS solution (1%, *w*/*v*) were prepared by dissolving in phosphate buffer (50 mM, pH 8.5, 50 mM NaCl, respectively. Then, the TNBS solution was mixed with the protein dispersion in equal volumes. 2 mL of the mixture was placed in a 60 °C water bath for 2 h. After cooling to room temperature, 1 mL of SDS (10%, *w*/*v*) and 0.5 mL of HCl (1 M) were added. The absorbance was recorded at 335 nm by a UV-VIS spectrophotometer (Nanodrop–2000C, Thermo Scientific, Waltham, MA, USA), where the absorbance of NOVA was set to 100%, and the degree of succinylation was defined as the percentage decrease in absorbance of the modified protein compared to NOVA.

### 2.4. Zeta-Potential Analysis 

The zeta potential of protein samples was determined using a Zetasizer Nano ZS (Malvern Instrument Ltd., Malvern, UK), where the protein solution was diluted to 1 mg/mL using deionized water and adjusting the pH to a range between 3.0 and 6.0. 

### 2.5. Sodium Dodecyl Sulfate-Polyacrylamide Gel Electrophoresis (SDS–PAGE) 

SDS–PAGE of proteins was carried out with 12% separating gels and 5% stacking gels. Firstly, protein samples (2 mg/mL) were diluted by a phosphate buffer (10 mM, pH 7.0). The protein samples were mixed with the loading buffer in a ratio of 4:1, and then 100 μL of the mixture was boiled in a water bath for 10 min. After cooling to room temperature, the mixture and protein markers (10–250 kDa) were added to the gel. The electrophoresis was carried out in a running buffer (25 mM Tris, 192 mM glycine, and 0.1% SDS, pH 8.3) at 80 V for 40 min and then at 120 V for 1 h. The gels were fixed in the fixing solution (ethanol: glacial acetic acid: distilled water, 5:1:4, *v*/*v*) for 20 min, then transferred to the staining solution (Coomassie Brilliant Blue R–250) for 1 h, and finally destained in the destaining solution (250 mL of 95% ethanol mixed with 80 mL glacial acetic acid to 1000 mL) until the background was clear.

### 2.6. Fourier Transform Infrared (FTIR) Spectroscopy

FTIR spectra of protein samples were measured using Fourier transform infrared spectrometer (Perkin-Elmer 16PC spectrometer, Boston, MA, USA). The lyophilized protein samples (1 mg) were mixed with KBr powder at a mass ratio of 1:100 and dried thoroughly, the mixture was then well milled and pressed into translucent slices. With air used as a background before each test, the wavenumber range, scanning times, and resolution were set to 400–4000 cm^−1^, 32, and 4 cm^−1^, respectively.

### 2.7. Circular Dichroism (CD) Analysis

The secondary structure of the proteins was analyzed using circular dichroism (Jasco 810, Jasco Corp., Tokyo, Japan) with reference to the previous method [13]. The protein samples were diluted to 0.1 mg/mL with phosphate buffer (10 mM, pH 7.0), where the phosphate buffer (10 mM, pH 7.0) was also used as the background. The optical length of the cuvette, the bandwidth, the resolution, and the scanning speed were set at 0.1 cm, 190–240 nm, 1 nm, 0.5 nm, and 100 nm/min, respectively. To reduce noise, each spectrum was obtained as the average of three scans. The secondary structure ratios of the proteins were predicted by Young’s equation using Protein Secondary Structure Estimation software of Spectra Manager (Spectra Manager Version 2, JASCO, Tokyo, Japan).

### 2.8. Intrinsic Fluorescence Emission Spectrum

The intrinsic fluorescence emission spectra of proteins were analyzed by a fluorescence spectrophotometer (RF-5300, Shimadzu, Japan) at 25 °C [14]. The protein sample solution (0.2 mg/mL) was prepared with phosphate buffer (10 mM, pH 7.0), where the phosphate buffer was set as blank. The optical length of the quartz cuvette, excitation wavelength, scan range of emission wavelength, and slit width was 1 cm, 290 nm, 300–400 nm, and 3 nm, respectively.

### 2.9. Free sulfhydryl Groups (SH) Determination

Determination of free sulfhydryl groups in protein samples was performed according to the Ellman method [15] with minor modifications. The protein dispersion (1 mg/mL) was prepared in Tris–Glycine buffer (0.086 mol/L Tris, 0.09 mol/L Glycine, 0.004 mol/L EDTA, pH 8.0). Next, the protein solution was mixed thoroughly with Ellman’s reagent (4 mg/mL DTNB in Tris–Glycine buffer, pH 8.0) at a volume ratio of 1:100 and reacted under dark conditions at room temperature for 1 h. The absorbance was recorded at 412 nm by a UV–VIS spectrophotometer (Nanodrop–2000C, Thermo Scientific, Waltham, MA, USA) using the Tris–Glycine buffer as blank. The content of free –SH is calculated as:(1)$$SH\left(\mathrm{umol}/g\;\mathrm{protein}\right)=73.53\times A_{412}\times D/C$$
where *A*_412_ is the absorbance of the sample at 412 nm, *D* is the dilution factor, and *C* is the concentration of the sample.

### 2.10. Surface Hydrophobicity (H_0_) Determination

H_0_ determination was carried out in accordance with the procedure of Sheng et al. [16]. The protein sample solution was diluted to different concentrations (0.20, 0.40, 0.60, 0.80, and 1.0 mg/mL) with phosphate buffer (10 mM, pH 7.0). Then 10 μL of ANS (8 mM, pH 7.0) was added to 2 mL of each diluted sample solution, mixed thoroughly with a vortex shaker, and placed in the dark for 15 min. The fluorescence intensity was measured using a spectrofluorophotometer (RF–5300, Shimadzu, Japan) at an emission wavelength of 470 nm and an excitation wavelength of 390 nm. When plotting the figure of fluorescence intensity vs. the concentration of the protein, the slope was defined as the H_0_ of the protein.

### 2.11. Differential Scanning Calorimetry (DSC)

Thermal analysis of proteins was carried out using DSC (CC 299 F1, NETZSCH Ltd., Selb, German). During each measurement, an appropriate amount of proteins, usually between 5 and 10 mg, was weighed into a closed aluminum crucible and tested between 30 and 160 °C at a rate of 10 °C/min. The aluminum crucible without the protein sample was used as blank.

### 2.12. Interfacial Tension Measurements

A K12 tensiometer (Krüss GmbH, Hamburg, Germany) was utilized to measure the interfacial tension between the water phase (3 mg/mL protein in deionized water, pH 7.0) and oil phase (soybean oil) or air phase was measured using a tensiometer using the Wilhelmy’s plate method [17]. Briefly, before each measurement, the platinum plate (length × width × height = 10 mm × 19.9 mm × 0.2 mm) was cleaned with deionized water, then burned to red, and cooled to room temperature before use. During the test of the air–water interfacial tension, the test was started by adjusting the height of the platinum plate to a height of 3 mm immersed in the water phase. During the oil–water interfacial tension test, the platinum plate was adjusted to be immersed in 20 mL of the water phase, and then 45 mL of oil phase was carefully added above the water phase to form the oil–water interface, and the test was started. The measurement was stopped when the standard deviation of the interfacial tension value was less than 0.05 mN/m. The test time was 3600 s between the water and oil phases and 1800 s between the water and air phases, and the temperature was maintained at 25 °C throughout the test.

### 2.13. Solubility

The solubility of protein was measured following the previous method [18] with some modifications. Sample solutions (2 mg/mL) were prepared by dissolving the proteins in phosphate buffer (10 mM, pH 7.0) and then centrifuged at 15,000 rpm for 20 min at 4 °C to get supernatants. The content of protein in the supernatant and in the original solution was measured following the Lowry method [19]. Triplicate determinations were carried out, and the solubility was obtained using the following equation:(2)$$\mathrm{Solubility}\;(\%)=\frac{Protein\;content\;in\;supernatant}{Total\;\;protein\;content}\times 100$$

### 2.14. Emulsifying Properties

Emulsifying properties were characterized by measuring the emulsifying activity index (EAI) and emulsion stability index (ESI) [20]. Firstly, protein samples (1%, *w*/*v*) were prepared in phosphate buffer (10 mM, pH 7.0). Then, the protein solution was mixed with soybean oil (4:1) and homogenized at 10,000 rpm for 2 min to form an emulsion. Then 50 μL of the emulsion was added to 5 mL of SDS solution (1 g/L) at 0 min and 10 min, respectively, where the SDS was also used as a blank. Finally, the absorbance was measured at 500 nm by a UV–VIS spectrophotometer (Nanodrop-2000C, Thermo Scientific, Waltham, MA, USA). EAI and ESI were calculated as follows:(3)$$EAI=\frac{2*2.303*A_{0}*N}{C*\varphi *10000}$$
(4)$$ESI=\frac{A_{10}}{A_{0}}*100$$
where *A*_0_ is the absorbance at 0 min, *N* is the dilution factor, *φ*, 0.25, is the volume fraction of oil in the solution, *C*, 0.01 g/mL, is the mass concentration of the original sample, and *A*_10_ is the absorbance of the emulsion at 10 min.

### 2.15. Foaming Properties

Foaming properties were characterized by measuring foaming capacity (FC) and foaming stability (FS). During each measurement, the protein samples (1%, *w*/*v*) were prepared in phosphate buffer (10 mM, pH 7.0). Then, 20 mL of the protein dispersion was placed in a measuring cylinder (internal diameter × height, 22 mm × 150 mm) and homogenized at 12,000 rpm for 2 min XHF–DY, Scientz, Zhejiang, China) equipped with a probe (internal diameter of 7.5 mm, height of 145 mm) to form foam. The foam volume at 1 min and 30 min was recorded, respectively. FC and FS were calculated by the following formula:(5)$$FC=\frac{V_{0}-V_{L}}{V_{L}}*100\%$$
(6)$$FS=\frac{V_{30}}{V_{0}}*100\%$$
where *V*_0_ is the volume of foam right after the stir, *V_L_* is the volume of the sample solution, and *V*_30_ is the volume of foam after 30 min of the stand.

### 2.16. Statistical Analysis

Each experiment was repeated at least three times. The results were displayed as the mean ± standard deviation. The difference between the means was considered significant at *p* < 0.05 (Duncan’s new multiple range tests). All the data were statistically analyzed using the SPSS software (version 12.0 for Windows; SPSS Inc., Chicago, IL, USA). 

## 3. Results and Discussion

### 3.1. Physicochemical Properties 

#### 3.1.1. Degree of Succinylation

Succinylation of proteins is a type of nucleophilic substitution reaction induced by the attack of a nucleophilic group (free amino, hydroxyl, or mercapto group) in the peptide chain on the carbonyl carbon atom of the succinic anhydride. Among the nucleophilic groups, the ε–amino group of lysine reacts preferentially [21], and its succinylation mechanism is shown in Figure 1A. It can be seen that the original free amino group of the protein is substituted by the succinyl group and therefore the extent of the succinylation is evaluated by monitoring the degree of reduction of the free amino group.

As shown in Figure 1B, the modification degree increased rapidly with the increasing addition of succinic anhydride. However, when the concentration of succinic anhydride reached 20% (*w/w*, protein basis), the increased rate of protein succinylation began to slow down, with a final percentage of succinylation determined at 69.37%. Similar results were reported in the soy protein isolate [22]. This may be because succinylation of OVA occurred gradually from the surface to the interior, with succinic anhydride first reacting with the amino groups on the surface of the protein at a rapid rate. When the amount of anhydride was further increased, the natural structure of the protein changed significantly, exposing the internal amino groups and changing the level of succinylation.

#### 3.1.2. FTIR

The changes in protein functional groups before and after the modification were characterized using FTIR [23]. As shown in Figure 1C, the shape of the OVA remained the same before and after the modification. The characteristic peaks at 1653 cm^−1^ and 1536 cm^−1^ were attributed to the amide I and amide II bands of the protein, respectively, where the amide I band corresponded to C=O stretching vibrations and C–N stretching vibrations, and the amide II band corresponded to N–H bonding and C–N stretching vibrations. The broad peaks at 3305 cm^−1^ were O–H and N–H stretching vibrations. With the enhancement of the succinylation, enhanced intensity of the peaks in the amide I and amide II bands can be observed, which was caused by the introduction of new O–H, C–N bonds, and C=O bonds by the succinyl group [24,25]. Notably, the peak at 1401 cm^−1^ (corresponding to the symmetric stretching vibration of the COO^−^ group) enhanced with the intensity of the succinylation, which was directly related to the extent of the succinylation [26]. The result showed that succinic anhydride was bound to OVA by covalent bonds.

#### 3.1.3. SDS–PAGE

The electrophoretic bands of natural and three different levels of acylation of OVA are shown in Figure 1D. In NOVA, the main band at 35–55 kDa was observed, which was the characteristic band of OVA (~45 kDa). Compared to NOVA, the electrophoretic band of SOVA shifted upwards and meanwhile became slightly lighter with succinylation, indicating that the succinylation could increase the molecular weight of OVA. As a result of the main stage of succinylation, the ε–amino group of the protein can be covalently bonded to the succinic anhydride, introducing more succinyl groups and increasing the molecular weight of the protein. This result is consistent with the study of succinylation–modified yolk proteins [27].

#### 3.1.4. DSC

The thermal stability of OVA was assessed by DSC. As shown in Figure 1E, the peak melting temperature (Tp) of NOVA was 73.44 ± 0.14 °C, which was consistent with the results reported in the literature [28]. Tp is positively related to the forces that maintain the conformation of a protein, so a higher Tp indicates better thermal stability [29]. With the enhancement of succinylation, the Tp of OVA showed a tendency to increase. Compared to NOVA, the Tp of SOVA–3 increased by about 14.9%, which meant that succinylation could significantly improve the thermal stability of OVA. This may be caused by the increased β–sheet content after the succinylation (Figure 2B), where proteins with a more ordered structure possessed higher thermal stability [30].

### 3.2. Structural Properties

#### 3.2.1. CD spectroscopy

The secondary structures of OVA with different succinylation degrees were evaluated by far–UV CD spectroscopy. As shown in Figure 2A, a negative band at ~208 nm and a negative shoulder peak at ~222 nm suggested the existence of α–helix structure. A strong positive band at ~195 nm and a negative minimum at ~217 nm indicated the abundant β–sheet content of the protein samples [16]. It was observed that there were dramatic changes in the CD spectrum intensity of the OVA upon succinylation.

The secondary structure was obtained from the far–UV CD processed by Young’s equation. As shown in the table inserted in Figure 2A, the main secondary structure of NOVA was composed of α–helix (28.5%), β–sheet (28.1%), β–turn (15.3%), and random coil (28.0%). With increasing succinylation, there was no significant change in the content of random coil structures (*p* > 0.05), while the content of α–helix and β–turn structures decreased continuously, accompanied by a sharp increase of β–sheet content (*p* < 0.05). Compared with the NOVA, the content of α–helix and β–turn structures of SOVA–3 decreased by 7.2% (from 28.5% to 21.3%) and 5.5% (from 15.3% to 9.8%), respectively, while β-sheet increased by 14.9% (from 28.1% to 43.0%). This indicated that the succinylation caused the conversion of the α–helix and β–turn to β–sheet in the secondary structure of the protein. This is similar to the results reported for rapeseed proteins after succinylation [31]. This may be due to the covalent binding of succinic anhydride to the protein, involving a condensation reaction between the carboxyl group and the ε–amino group, which was located in the α–helix structure or its neighboring region [21]. According to Feng et al. [32], the increased content of β–sheet facilitated the hydrogen bonding between protein molecules, thus improving the thermodynamic stability of the protein, as evidenced by the results of DSC measurements (Figure 1E).

#### 3.2.2. Intrinsic Fluorescence

Intrinsic fluorescence can detect the aqueous environment surrounding tryptophan residues by monitoring the fluorescence intensity and maximum emission wavelength (λ_max_), providing vital information on the tertiary structure conformation of the protein [33]. The results of the succinylation of OVA are shown in Figure 2A. As succinylation increased, the fluorescence intensity of OVA gradually decreased. This was attributed to the shielding effect of the introduction of the long succinyl chain, which enhanced the spatial barrier within the protein so that some of the Trp residues were encapsulated within the confined space [34]. In addition, the λ_max_ of the protein underwent a positional change with increasing succinylation. Compared to NOVA, the λ_max_ of SOVA–3 gradually increased from 336 nm to 340 nm, performing a red–shift phenomenon. This can be attributed to the further unfolding of the protein structure after succinylation, which exposed the buried internal Trp residues to a more polar environment [35].

#### 3.2.3. Surface Hydrophobicity (H_0_)

The H_0_ of the protein was assessed using the fluorescence intensity of ANS as a probe that binds to hydrophobic groups on the protein surface. The results are shown in Figure 2C, where a significant decrease in H_0_ of OVA can be observed with increasing succinylation, similar to the results of the oat protein isolate [12] and the whey protein [18] modified by succinylation. This can be explained by the fact that, on the one hand, succinylation can further stretch the structure of the protein, exposing the inside hydrophobic groups and resulting in an increased H_0_; on the other hand, the free amino group on the surface of the protein was replaced by a more hydrophilic carboxyl group after modification, which resulted in a decrease in H_0_ [36]. In addition, the increased electronegativity of the succinylation-modified protein may inhibit the binding of ANS^−^ anion to hydrophobic groups on the surface of the molecule, which can also partially explain the decrease in H_0_. 

#### 3.2.4. Free Sulfhydryl Groups

According to Figure 2D, the free sulfhydryl content of OVA increased significantly with the enhancement of succinylation. Compared to NOVA, the free sulfhydryl content of SOVA–3 increased from 7.8 μmol/g to 49.7 μmol/g. This was due to the succinylation causing the protein structure to unfold, allowing the exposure of the –SH group inside the molecule. This is consistent with the results of the intrinsic fluorescence (Figure 2B).

The above findings indicated that the succinylation modification increased the electronegativity of the protein by introducing succinyl groups into the protein, causing further unfolding of the protein structure, resulting in changes in the conformation (including secondary and tertiary structures) and modifying the hydrophobic properties of the protein, which may have a significant impact on the functional properties of the protein, including solubility, emulsification, and foaming.

### 3.3. Interface Properties

#### 3.3.1. Oil–Water Interfacial Tension

As amphiphilic molecules, proteins reduce interfacial tension by adsorbing to the oil–water interface and forming a physical barrier. Generally, the lower the interfacial tension, the better the emulsification activity of the protein. Figure 3A shows the dynamic interfacial tensions at the oil–water interface for NOVA and SOVA. The initial interfacial tension of NOVA (26.63 mN/m) was significantly higher than that of the SOVA group. With enhanced succinylation, the initial interfacial tension of SOVA–3 decreased to 8.90 mN/m. This finding may be related to the conformational changes in the protein caused by succinylation. In this study, succinylation resulted in a higher content of β–sheet structures of OVA (Figure 2A) and a more extended structure (Figure 2B, D). The more stretched molecular structure facilitated the diffusion and adsorption of proteins at the oil–water interface [37]. Furthermore, the higher content of β–sheet structures enhanced protein interactions at the oil–water interface and promoted the formation of denser and more rigid interfacial films [38]. Thus, succinylation effectively reduced the interfacial tension at the oil–water interface.

#### 3.3.2. Air–Water Interfacial Tension

In general, the performance of a foaming agent can be estimated by its ability to reduce the surface tension at the air–water interface. As shown in Figure 3B, the surface tension of all proteins tended to decrease rapidly during the first 600 s of adsorption time (e.g., from 55.79 mN/m to 52.23 mN/m for NOVA), indicating the rapid diffusion of protein molecules and their adsorption to the air–water interface, but as the adsorption time increased, the decrease in surface tension became smaller, indicating the amount of protein adsorbed at the air–water interface gradually reached a maximum and formed a saturation state. The succinylation effectively reduced the surface tension, and SOVA–3, with the highest degree of succinylation, had the lowest surface tension. In general, the reduction in surface tension facilitated the rapid diffusion of protein molecules to the air–water interface and was more conducive to the adsorption, stretching, and rearrangement of protein molecules at the interface [39]. It was also reported that succinylation could decrease the surface tension of date palm pollen protein concentrate [40].

### 3.4. Functional Properties

#### 3.4.1. Protein Solubility

The effect of succinylation on the zeta potential of OVA at different pH is shown in Figure 4A. With greater degree of succinylation, the zeta potential of the protein at the same pH dropped and the isoelectric point (*PI*) of OVA kept moving towards a lower pH. Compared to NOVA, the *PI* of SOVA–3 dropped from around 4.2 to around 3.5. This was due to the partial replacement of the cationic ε–amino group (especially the lysine residue) by the anionic succinyl group, increasing the electronegativity of the protein and shifting its *pI* to lower values. A similar phenomenon was observed in acylated kidney protein isolate [41]. It was demonstrated that the electronegativity of OVA could be enhanced by the introduction of a succinyl group.

Solubility, a key functional property of proteins, is strongly influenced by the polarity and charge density of the protein [18]. As seen in Figure 4B, succinylation can significantly enhance the solubility of OVA. Compared to NOVA, the solubility of SOVA–3 was increased by 21.5% (from 73.9% to 95.4%). This was because, on the one hand, the replacement of the original free amino group by a more hydrophilic terminal carboxyl group promoted protein–water interactions; on the other hand, the enhancement of the electrical potential on the protein surface prevented denaturation aggregation of the protein (Figure 4A), thus facilitating its stability in aqueous solution, and also improving its water solubility.

#### 3.4.2. Emulsifying Properties

EAI and ESI were used to assess the adsorption efficiency and adsorption stability of proteins at the oil–water interface, respectively. The results of the emulsification performance are shown in Figure 5A, where EAI and ESI of OVA were both significantly enhanced (*p* < 0.05) as the succinylation increased. Compared to NOVA, the EAI and ESI of SOVA–3 increased by 1.6 times (from 23.85 m^2^/g to 61.68 m^2^/g) and 1.2 times (from 23.42% to 50.69%), respectively. It was because that succinylation could effectively lower their interfacial tension at the oil–water interface (Figure 3A). In addition, the introduction of succinyl groups into the protein increased the electronegativity of OVA, imparting a stronger potential to the ionized layer around the oil droplets in the emulsion, which could effectively prevent aggregation among the droplets and thus improve the ESI of the emulsion [42]. As can be seen from Figure 5B, the emulsion formed by NOVA at 0 d was significantly stratified and had poor emulsification properties, whereas the emulsion formed by the experimental group was more homogeneous. After 1 d of storage, the break-up of the emulsion was observed in the experimental group and the height of the emulsion layer increased significantly with the degree of modification. Among them, emulsions formed by SOVA–3 were the most stable, and no significant separation was observed. Similar findings have been reported for acylation–modified soy protein [43]. This suggests that succinylation modification can significantly improve the emulsification properties of OVA.

#### 3.4.3. Foaming Properties

Similarly, the foaming properties of proteins depend on the efficiency of protein migration to the air–water interface to form an interfacial film and the ability to maintain interfacial film stability [44]. As can be seen from Figure 6A, succinylation significantly enhanced the FA and FS of OVA. 2.7 times (from 12.2% to 44.6%) and 1.5 times (from 35.1% to 87.6%) the FA and FS of SOVA–3, respectively, compared to NOVA. Additionally, after 1 d of storage, the foam produced by SOVA–3 showed better stability (Figure 6B). This is consistent with the changes in the EAI and ESI measurements, which can be equally explained by the adsorption behavior of the protein at the interface as described above. 

The above results indicated that the succinylation helped to enhance the functional properties of the proteins, including solubility, emulsification properties, and foaming properties.

## 4. Conclusions

Succinylation improved the functional properties by modulating the charge density and conformation of OVA. By introducing succinyl groups into the proteins, both the electronegativity and the hydrophilicity of the protein were enhanced, resulting in a significant improvement in the solubility of OVA. The succinylation transformed the α–helix and β–turn into a thermostable β–sheet and stretched the tertiary structure of the protein, and further exposed active groups. The change in protein conformation reduced the interfacial tension (oil–water and air–water interface) of SOVA, resulting in a significant improvement in emulsification and foaming properties. These results suggest that succinylation has great potential for modulating the relationship between conformational and functional properties of protein products.

## Figures and Tables

**Figure 1 foods-11-02724-f001:**
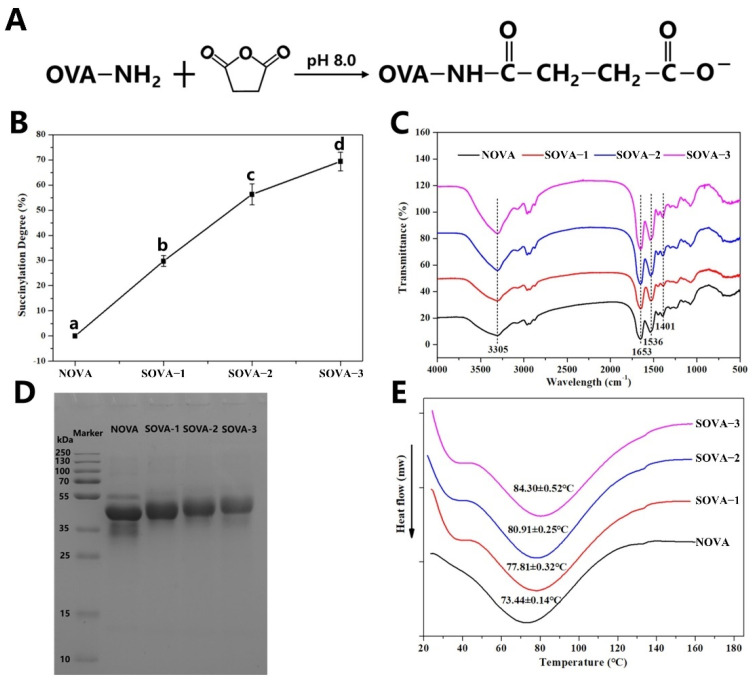
Nucleophilic substitution mechanism of OVA and succinic anhydride (**A**), succinylation degree (**B**), FTIR spectroscopy(**C**), SDS–PAGE patterns (**D**), and DSC thermograms (**E**) of OVA with a different succinylated degree. Different letters indicate significant differences (*p* < 0.05).

**Figure 2 foods-11-02724-f002:**
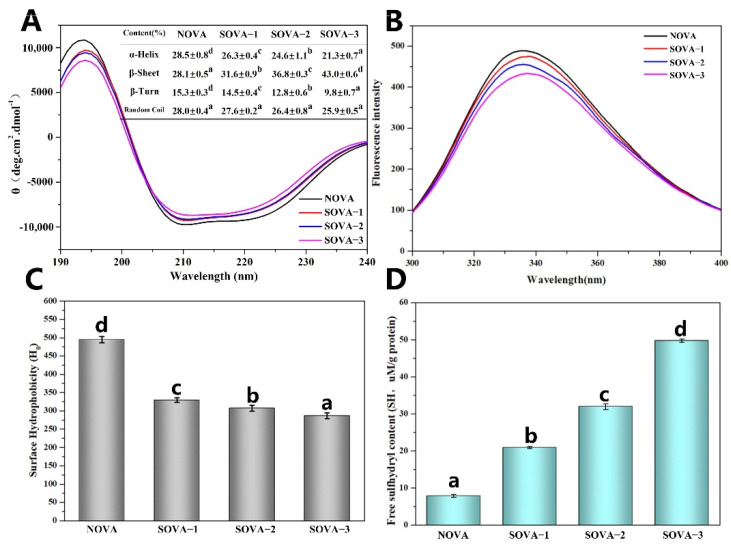
CD spectra (**A**), intrinsic fluorescence (**B**), surface hydrophobicity index (H_0_) (**C**), and free sulfhydryl (SH) content (**D**) of OVA with different succinylated degrees. Different letters above the bars indicate significant differences (*p* < 0.05).

**Figure 3 foods-11-02724-f003:**
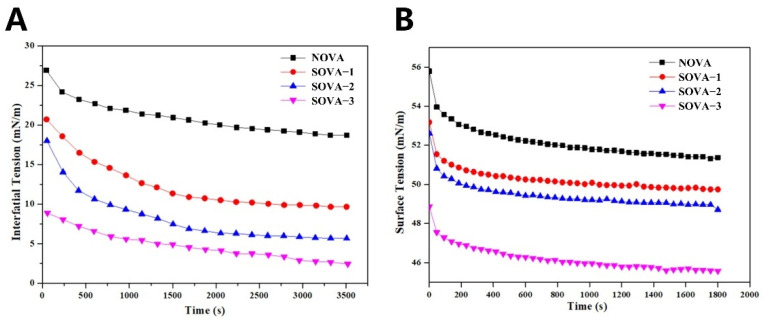
Interface tension between oil and protein solution (**A**), and surface tension between air and protein solution (**B**) of OVA with different succinylated degrees.

**Figure 4 foods-11-02724-f004:**
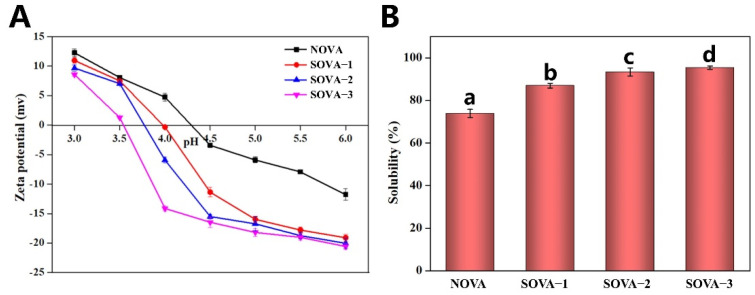
Zeta potential (**A**) and protein solubility (**B**) of OVA with different succinylated degrees. Different letters above the bars indicate significant differences (*p* < 0.05).

**Figure 5 foods-11-02724-f005:**
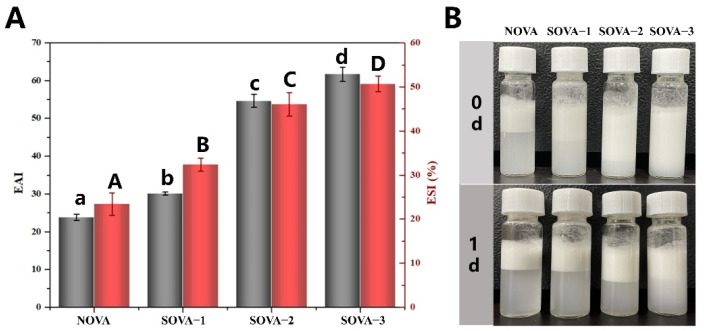
Emulsifying properties (**A**) and emulsion visual appearance stored for 0 d and 1 d (**B**) stabilized by OVA with different succinylated degrees. Different letters above the bars indicate a significant difference in the same indicator (*p* < 0.05).

**Figure 6 foods-11-02724-f006:**
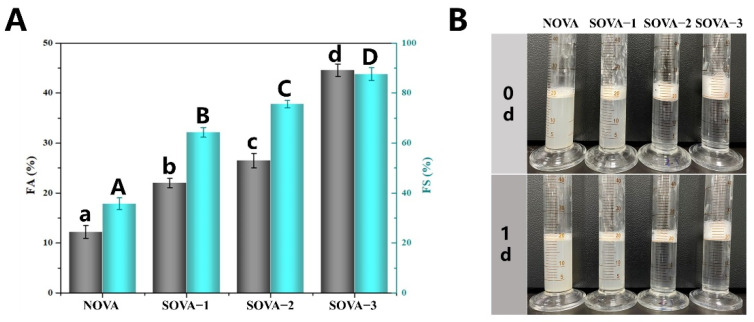
Foaming properties (**A**) and foam visual appearance stored for 0 d and 1 d (**B**) stabilized by OVA with different succinylated degrees. Different letters above the bars indicate a significant difference in the same indicator (*p* < 0.05).

## Data Availability

Data is contained within the article.

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
