# Peer review of "Succinylation Modified Ovalbumin: Structural, Interfacial, and Functional Properties"

_foods, 2022, doi:10.3390/foods11182724_

Round 1
Reviewer 1 Report
In the last sentence of the Abstract, the last phrase is repeated with 2 different sets of values for changes in the emulsifying activity and stability.
In Fig. 1, the labels for C and D are reversed.
Paragraph 3.2.3, in the last sentence, the word "comprehensive" does not make sense. Perhaps you meant "contradictory".
In the first paragraph of the Introduction, it says ovalbumin has important applications in "improving taste and texture of food products". Is there a taste test for modified proteins added to food products to determine if they adversely affect the taste of the food?
Author Response
Dear Reviewer 1:
Thanks for your kind suggestions to improve our manuscript “Succinylation modified ovalbumin: structural, interfacial, and functional properties” (foods-1886218). The manuscript has been revised according to your constructive and valuable comments. We have carefully answered the questions accordingly. We hope that our paper has met the journal requirements and format. If need further changes, we can modify for your convenience.
The main corrections in the paper and the responds to your comments are as flowing:
1.In the last sentence of the Abstract, the last phrase is repeated with 2 different sets of values for changes in the emulsifying activity and stability.
Response: Thanks for your nice note. We have corrected “emulsifying activity and stability … 2.7 times and 1.5 times, respectively” as “foaming capacity and stability… 2.7 times and 1.5 times, respectively”.
2.In Fig. 1, the labels for C and D are reversed.
Response: Thanks for your careful correction. We have reordered the labels for C and D.
3.Paragraph 3.2.3, in the last sentence, the word "comprehensive" does not make sense. Perhaps you meant "contradictory".
Response: Thanks for your careful correction. We have corrected "comprehensive" as "contradictory" as you suggested.
4.In the first paragraph of the Introduction, it says ovalbumin has important applications in "improving taste and texture of food products". Is there a taste test for modified proteins added to food products to determine if they adversely affect the taste of the food?
Response: Thanks for your valuable question. What we want to emphasize in "improving taste and texture of food products" is the role of ovalbumin as a carrier (emulsion, foam, and gel) of flavor substances, which can improve the taste of food products by controlling the retention and release of flavor substances in food products [1]. As you mentioned, studying the effect of modified proteins on food flavor is also a very interesting subject. There are some important researches in recent years. Nasiru et al. [2] studied the effect of high-voltage cold plasma (HVCP) on the egg white proteins (EWP)’ flavour attributes, and found an overall increase in the concentrations of the volatile compounds of EWP as the HVCP treatment time increased.; Chang et al. [3] investigated the effects of microwave treatment-induced Maillard modification of egg white protein on the baking performance of cake paste, and found the modification didn’t affect the flavor of cake paste; Nasiru et al. [4] explored the effect of glycation on volatile flavor characteristics of silver carp mince proteins, and found glycation could enhance the volatile flavor of silver carp mince during thermal processing.
[1] Liu, T., Zhao, Y., Wu, N., Chen, S., Xu, M., Du, H., ... & Tu, Y. (2022). Egg white protein-based delivery system for bioactive substances: a review. Critical Reviews in
Food Science and Nutrition, 1-21.
[2] Nasiru, M. M., Umair, M., Boateng, E. F., Alnadari, F., Khan, K. U. R., Wang, Z., ... & Korma, S. A. (2022). Characterisation of Flavour Attributes in Egg White Protein Using HS-GC-IMS Combined with E-Nose and E-Tongue: Effect of High-Voltage Cold Plasma Treatment Time. Molecules, 27(3), 601.
[3] Chang, C., Su, Y., Gu, L., Li, J., & Yang, Y. (2021). Microwave induced glycosylation of egg white protein: study on physicochemical properties and baking performance. Food Hydrocolloids, 118, 106569.
[4] Liu, J., Shen, S., Xiao, N., Jiang, Q., & Shi, W. (2022). Effect of glycation on physicochemical properties and volatile flavor characteristics of silver carp mince. Food Chemistry, 386, 132741.
Reviewer 2 Report
Title: Succinylation modified ovalbumin: structural, interfacial, and functional properties
This manuscript adds more valuable and interesting research findings regarding the structure of ovalbumin and changes as per happened by addition of succinic anhydride. The manuscript is very well presented, findings are sound and discussed reasonably. A few general issues require the attention of authors:
The quality of the graphs and the graphic should be improved (presented dada tables in the graphs, and data labels), the SDS-PAGE protein bands require to be labelled and the bands should be identified along with standard marker.
Abstract: Please modify the first sentence as: “With addition of different…”
Before “respectively” please place a comma.
Introduction: The Introduction Is well presented by specifying the main background findings and related issues.
Materials and Methods:
The M & M are well presented in details. However, all the equations within manuscript requires a numbering system.
Author Response
Dear Reviewer 2:
Thanks for your kind suggestions to improve our manuscript “Succinylation modified ovalbumin: structural, interfacial, and functional properties” (foods-1886218). The manuscript has been revised according to your constructive and valuable comments. We have carefully answered the questions accordingly. We hope that our paper has met the journal requirements and format. If need further changes, we can modify for your convenience.
The main corrections in the paper and the responds to your comments are as flowing:
This manuscript adds more valuable and interesting research findings regarding the structure of ovalbumin and changes as per happened by addition of succinic anhydride. The manuscript is very well presented, findings are sound and discussed reasonably.
Response: Dear Reviewer 2, many thanks for your valuable comments and suggestions regarding to our manuscript.
1.The quality of the graphs and the graphic should be improved (presented dada tables in the graphs, and data labels), the SDS-PAGE protein bands require to be labelled and the bands should be identified along with standard marker.
Response: Thanks for your kind suggestion. We have made careful changes to the images based on your suggestions to improve the quality of the graphs, including increasing the resolution of the images and the size of the text (Figure 1, Figure 2, and Figure 4).
2.Abstract: Please modify the first sentence as: “With addition of different…”
Response: Thanks for your nice note. We have modified the sentence as “with addition of different levels…” as you suggested.
3.Before “respectively” please place a comma.
Response: Thanks for your careful correction. We have placed a comma before “respectively” as you suggested.
4.Introduction: The Introduction Is well presented by specifying the main background findings and related issues.
Response: Thanks for your attention.
5.Materials and Methods:
The M & M are well presented in details. However, all the equations within manuscript requires a numbering system.
Response: Thanks for your kind suggestion. We have numbered all the formulas as you suggested.